# Phytochemical Compounds Involved in the Bone Regeneration Process and Their Innovative Administration: A Systematic Review

**DOI:** 10.3390/plants12102055

**Published:** 2023-05-22

**Authors:** Alina Hanga-Farcaș, Florina Miere (Groza), Gabriela Adriana Filip, Simona Clichici, Luminita Fritea, Laura Grațiela Vicaș, Eleonora Marian, Annamaria Pallag, Tunde Jurca, Sanda Monica Filip, Mariana Eugenia Muresan

**Affiliations:** 1Doctoral School of Biomedical Science, University of Oradea, 410087 Oradea, Romania; alinushanga@yahoo.com; 2Department of Preclinical Discipline, Faculty of Medicine and Pharmacy, University of Oradea, 10, 1 December Square, 410073 Oradea, Romania; florinamiere@uoradea.ro (F.M.); fritea_luminita@yahoo.com (L.F.); marianamur2002@yahoo.com (M.E.M.); 3Department of Physiology, Iuliu Hațieganu University of Medicine and Pharmacy, 8 Victor Babeș Street, 400347 Cluj-Napoca, Romania; gabriela.filip@umfcluj.ro (G.A.F.); sclichici@umfcluj.ro (S.C.); 4Department of Pharmacy, Faculty of Medicine and Pharmacy, University of Oradea, 10, 1 December Square, 410073 Oradea, Romania; marian_eleonora@yahoo.com (E.M.); annamariapallag@gmail.com (A.P.); jurcatunde@yahoo.com (T.J.); 5Department of Physics, Faculty of Informatics and Sciences, University of Oradea, 1 University Street, 410087 Oradea, Romania; sfilip@uoradea.ro

**Keywords:** bone regeneration, bone cells, bone biomarkers, bone signaling pathways, plant extracts, phytochemicals compounds, nanometric systems, nanostructured scaffolds

## Abstract

Bone metabolism is a complex process which is influenced by the activity of bone cells (e.g., osteocytes, osteoblasts, osteoclasts); the effect of some specific biomarkers (e.g., parathyroid hormone, vitamin D, alkaline phosphatase, osteocalcin, osteopontin, osteoprotegerin, osterix, RANKL, Runx2); and the characteristic signaling pathways (e.g., RANKL/RANK, Wnt/β, Notch, BMP, SMAD). Some phytochemical compounds—such as flavonoids, tannins, polyphenols, anthocyanins, terpenoids, polysaccharides, alkaloids and others—presented a beneficial and stimulating effect in the bone regeneration process due to the pro-estrogenic activity, the antioxidant and the anti-inflammatory effect and modulation of bone signaling pathways. Lately, nanomedicine has emerged as an innovative concept for new treatments in bone-related pathologies envisaged through the incorporation of medicinal substances in nanometric systems for oral or local administration, as well as in nanostructured scaffolds with huge potential in bone tissue engineering.

## 1. Introduction

Bone is a complex tissue which has multiple functions and which presents a unique structure that is in permanent renewal. Bone regeneration maintains the integrity of the tissue and is coordinated by two processes: bone resorption carried out by osteoclasts, through which the removal of aged cells takes place, and the formation of new bone tissue by osteoblasts. The two processes are balanced [1]. The bone remodeling process is a continuous and cyclic one that takes place at the level of a temporary anatomical structure called the Basic Multicellular Unit (BMU) [2]. Bone remodeling at the BMU level takes place in three phases: (i) The Initiation Phase involves the selection of osteoclastic precursors and their differentiation into mature osteoclast cells and their activation for bone resorption; (ii) The Inversion Phase involves the inhibition of osteoclast activity and their apoptosis, as well as the differentiation of osteoblasts; (iii) The Terminal Phase involves the formation of new bone tissue by osteoblasts [3].

Bone formation or osteogenesis performed by the osteoblast and bone resorption or osteolysis achieved by the osteoclast are the main processes that maintain bone homeostasis. The balance between these two processes is regulated by hormonal and signaling pathways.

Parathyroid hormone (PTH) is one of the most important hormonal bone metabolism regulators. The normal activity of PTH ensures the reabsorption of calcium at the bone level. PTH binds to its receptor, type 1PTH receptor (PTHR1), which leads to the activation of protein kinase A (PKA) or protein kinase C (PKC). Using these signaling pathways, PTH can determine both bone formation and bone resorption [4]. PTHR1 is found on the surface of several cells: osteoblasts, osteoclasts and renal tubule cells; its effect on osteoblasts is a decrease in cell apoptosis, and on osteoclasts a decrease in sclerostin production [5,6]. The blood levels of PTH are directly influenced by the Ca^2+^ and vitamin D concentrations. Low levels of vitamin D cause hyperparathyroidism [7]. Low levels of vitamin D lead to rickets in children and osteomalacia in adults. Moreover, the use of vitamin D supplements has been correlated with an increase in calcium levels and bone density and a reduction in bone resorption and the risk of bone fractures [8,9].

In order to better understand bone metabolism, we will review the bone cells’ biomarkers involved in bone metabolism and bone regeneration and their main regulatory pathways. Additionally, the biomaterials used in bone regeneration; the plant extracts and phytochemical compounds demonstrating a positive effect on the bone regeneration process; and the pathogenetic mechanisms involved will all be described.

### 1.1. Bone Metabolism

#### 1.1.1. The Main Cells of Bone Metabolism

Three main cell types are involved in the bone tissue constitution: the osteoblast, responsible for new bone formation; the osteoclast, responsible for osteolysis and osteocytes; and the mature osteoblasts, located in the bone matrix and responsible for coordinating bone hemostasis.

*Osteocytes:* Osteocytes are cells located at the level of the bone matrix and which are derived from mature osteoblasts [10]. They represent approximately 90–95% of bone cells, have a long life of up to 25 years, are made up of the cell body and dendrites and are found at the level of the lacuno-canalicular system, which allows their connection to adjacent osteocytes, osteoblasts, osteoclasts, bone marrow, blood vessels and nerves [11]. Osteocytes play a central role in bone homeostasis by acting as mechanical sensors and mechanical transducers for the different stimuli to which the bone is subjected. They control through chemical and hormonal responses both the process of osteolysis, by activating the cellular differentiation process of osteoclasts, and that of osteogenesis, by forming young osteocytes from osteoblasts [12,13].

*Osteoblasts:* Osteoblasts are small, mononucleated, cubic-shaped cells, but they can also appear in a flat or cylindrical form, and they are derived from the mesenchymal cells located in the bone marrow. Their differentiation takes place through the action of a network of cytokines and transcription factors [14]. The mesenchymal cells located at the level of the bone marrow have a multipotent character from their level and derive several cell lines: adipocytes, chondrocytes and osteoblasts [15]. Osteoblastogenesis is the cellular differentiation process of osteoblasts from mesenchymal cells. This process takes place in three phases: proliferation, matrix maturation and mineralization. The three phases are characterized by the expression of specific osteoblastic genetic markers such as: osteocalcin (OCN), osteopontin (OPN), bone sialoprotein (BSP), collagen type I (COL1A) and alkaline phosphatase (ALP). Osteoblastogenesis is controlled by hormonal factors (parathormone and glucocorticoids); vitamin D3; specific signaling pathways (Wnt, BMP, Hedgehog, Notch); circulating cytokines and transcription factors (Runx2 and Osx1); fibroblast growth factor (FGF); and transforming growth factor β (TGFβ) [16].

*Osteoclasts:* Osteoclasts are the cells responsible for bone resorption. They are large, multinucleated ones that are derived from hematopoietic cells, formed by the fusion of mononuclear precursors of the monocyte-macrophage line. The cellular differentiation process requires the coordination of transcription factors with co-activating and co-repressing factors [17]. In the process of cellular differentiation and maturation of osteoclasts, a central role is played by the activator receptor of nuclear factor kB ligand (RANKL). Secreted by osteoblasts, bone marrow cells and lymphocytes, RANKL is part of the tumor necrosis factor family. RANKL interacts with the RANK receptor on the cells of preosteoclasts and thus promotes their differentiation into osteoresorbing mature osteoclasts [18].

#### 1.1.2. The Main Biomarkers of Bone Metabolism

Bone metabolism represents the sum of the biochemical processes through which osteolysis and osteogenesis take place, and it represents both the metabolism of proteins and of the minerals that enter the composition of the bone. The regulation of bone metabolism is done by a series of hormonal or enzymatic biomarkers.

*Parathyroid hormone* (PTH): PTH influences the biochemical processes that take place in bone remodeling. Low PTH values cause a decrease in bone circulatory markers—both osteoforming and bone resorption markers—which causes a decrease in bone turnover. By contrast, in hypoparathyroidism bone mineral density increases. The treatment of hypoparathyroidism with human parathyroid hormone (hPTH) or recombinant human parathyroid hormone (rhPTH) increases bone turnover, while bone mineral density shows increases at the hip and spine level and decreases at the radiocarpal level [19].

*Vitamin D*: Vitamin D deficiency causes rickets in children and osteomalacia in adults, and it is associated with the occurrence of osteoporosis and the increased risk of fractures. Recent studies have demonstrated the effect of 25-hydroxy vitamin D3 and its metabolite 1α,25 hydroxy vitamin D3 to differentiate mesenchymal stem cells of human origin towards osteoblasts [20].

*Alkaline phosphatase*: The bone isozyme is a glycoprotein attached to the cell membrane of osteoblasts, from where it is released into circulation, and it is the main indicator of osteogenesis. It participates in the synthesis of hydroxyapatite, providing inorganic phosphate, pyrophosphate and monophosphoesters, and at the same time it hydrolyzes pyrophosphate, which is an inhibitor of the mineralization process [21]. Tissue-nonspecific alkaline phosphatase (TNSALP) is an enzyme essential in the bone mineralization process. The mutation of the ALP gene is responsible for the synthesis of the TNSAPL enzyme and leads to hypophosphatasia [22].

*Osteocalcin*: Osteocalcin is the most abundant non-collagenous protein at the bone level. It is dependent on vitamin K and is produced by osteoblasts. It presents in its structure three gamma-carboxyglutamic acids that have an affinity for the Ca^2+^ ion, which will influence the bone remodeling and mineralization processes. It acts simultaneously as an inhibitor of the hydroxyapatite growth and as a regulator of the activity of osteoclastic precursors [23,24].

*Osteopontin*: Osteopontin is a phosphoprotein secreted at the bone level—especially by osteoblasts—with a role in bone metabolism, in which it participates through endocrine, neurological and immunological processes. It acts as a parathormone regulator, with low osteopontin values blocking the PTH activity of stimulating alkaline phosphatase and osteocalcin expression [25].

*RANKL:* RANKL is the receptor activator of NF-kB (RANK) ligand; it is a homotrimeric protein secreted by osteoblasts with functions in osteoclastogenesis [26].

*Osteoprotegerin* (OPG): OPG is a glycoprotein secreted specifically by osteoblasts and is a cytokine receptor of the Tumor Necrosis Factor (TNF). OPG acts as a “bait” receptor for RANKL, inhibiting osteoclastogenesis and bone resorption [27].

*Osterix*: Osterix is a protein with a role in the differentiation of mesenchymal cells into osteoblasts, which inhibits the formation of chondrocytes. At the osteoblastic level, osterix indicates the genetic expression of osteopotin, osteonectin, type 1a1 collagen and bone sialoprotein, which are all necessary for the process of bone mineralization at the level of osteoblasts [28].

*Runx2*: Runx2 is an essential protein in the maturation process of osteoblasts. Its expression is weak in mesenchymal cells and strong in immature osteoblasts. Moreover, it is essential in the differentiation of mesenchymal cells into osteoblasts [29].

#### 1.1.3. The Main Signaling Pathways Specific to Bone Metabolism

Together with the hormonal regulation of bone homeostasis, an important role in this process belongs to the signaling pathways., as they are involved in the processes of embryonic bone development and bone repair [30].

*RANKL/RANK/OPG:* RANKL is produced by osteoblasts, and its binding to RANK at the level of osteoclastic precursor cells determines the differentiation of this cell line, favoring osteoclastogenesis. OPG secreted by osteoblasts is a “bait” receptor for RANKL, preventing its binding to RANK and thereby preventing osteoclastogenesis and bone resorption [26].

*Wnt/β signaling:* The Wnt pathway stimulates osteoblast activation and differentiation. Canonical Wnt causes β-catenin translocation and stabilization in the cell nucleus, which regulates gene transcription in response to Wnt signaling [31].

*Notch*: At the bone level, Notch receptors and their ligands are responsible for a multitude of phenomena: osteoblastic differentiation, bone matrix mineralization, osteoclast recruitment, cell fusion and osteoblast/osteoclast cell proliferation [32].

*Bone morphogenetic proteins* (BMPs): Part of the transforming growth factor β (TGFβ) family, they are involved in the processes of bone formation through the differentiation of osteoblasts. A central role in the BMP pathway signaling is played by SMAD 1, 5 and 8 proteins that interact with BMP receptors [33,34].

### 1.2. Biomaterials for Bone Regeneration

In many decades, a wide range of biomaterials have been developed to address bone defects promoting bone regeneration through various mechanisms, such as mechanical support, osteoconduction, osteoinduction, vascularization, neurotization, antibacterial effect, etc. [35]. Autologous bone grafts were considered the “gold standard” material for bone defects, but due to some disadvantages (limited quantity, long surgical procedures and morbidity), they have been replaced by synthetic bone grafts. Therefore, a plethora of emerging biomaterials possessing specific advantageous characteristics have been designed [36]. According to their dimensional structure and dimension size (as nanoscale), the biomaterials have been classified in zero-, one-, two-, three- and four-dimensional biomaterials [35]. Often, different classes of biomaterials have been combined, leading to hybrid composite biomaterials with synergic properties for bone tissue regeneration.

*Zero-dimensional biomaterials* have all three dimensions confined at nanoscale with a high surface-to-volume ratio, including some carbon-based nanomaterials (fullerene, nanodiamonds, carbon dots) and inorganic nanoparticles (NPs) (AuNPs, AgNPs, iron oxide NPs, etc.). They have presented great benefits for bone regeneration, such as biomineralization, osteogenic differentiation, good mechanical performance and biocompatibility [35].

*One-dimensional biomaterials* (two dimensions are nanosized) presenting high length-to-diameter ratio and unique nanotopography refer to nanowires (silicon nanowire—SiNW) and nanotubes (Titanium oxide nanotubes—TiO_2_NTs, carbon nanotubes—CNTs). They have modulated osteogenic and chondrogenic cell adhesion, proliferation and differentiation, and have also facilitated mineralization and demonstrated exceptional bone tissue compatibility [35].

*Two-dimensional biomaterials* (one dimension is in the nanoscale range), being characterized by a high diameter-to-thickness ratio, include graphene and its derivatives (graphene oxide, reduced graphene oxide). Their exceptional osteoinductive properties (in vitro and in vivo), along with their enhanced mechanical properties and favorable biocompatibility and facilitated mineralization, have been attributed to graphene due to various interactions with biomolecules and physical stress (affecting cytoskeletal tension and inducing cytoskeletal reorganization). Other nanofilm coatings (calcium phosphate coatings, black phosphorus (BP) nanofilms) have been applied to facilitate the integration of biomaterial [35].

*Three-dimensional biomaterials* (all dimensions are larger than the nanoscale) are the most widely used implants in clinics and include metallic (titanium and its alloy, silver, magnesium, niobium, strontium, stainless steel, cobalt, tantalum) [37]; bioceramic (bioactive bioceramics: calcium phosphate ceramics, hydroxyapatite, bioglass; bioinert bioceramics: alumina, zirconia, silicon carbide) [38]; and polymeric scaffolds and hydrogels. The disadvantages of metal-based scaffolds (poor biodegradability, local/systemic toxicity, higher elastic modulus) have been overcome by fabricating scaffolds with tunable porosity [35].

In the group of bioceramics, calcium phosphates and bioactive glass are the most frequently used in the orthopedic and dental fields. Calcium phosphates (with various forms, tunable porosities and densities) resemble the native bone tissue, having a good capacity for integration and great bioactivity and osteoconductivity [35]. Hydroxyapatite (HA) (one example of calcium phosphate) is an intrinsic component of bone tissue, presenting excellent biocompatibility, osseointegration, osteoconductivity, osteoinductivity and angiogenic effects. Biodegradable polymers have been mixed with HA in order to overcome its limitations (brittleness and insufficient mechanical strength). Bioactive glass (BG)—with its main components of silicate, borate and phosphate glass—exhibits ideal surface reactivity, bioactivity and osteoinductivity, but also brittleness (a disadvantage solved via polymer inclusion) [35].

Polymeric scaffolds are based on natural polymers (collagen, chitosan, hyaluronic acid, silk, alginate, gelatin, cellulose, etc.) and synthetic polymers (polylactic acid (PLA), poly(glycolic acid) (PGA), poly(lactic-co-glycolic acid) (PLGA), poly(ethylene glycol) (PEG), polycaprolactone (PCL), polyurethane (PU) etc.) [39,40]. Natural polymers and synthetic polymers have advantages, such as biocompatibility, design flexibility, supporting cell attachment, osteogenic differentiation, calcium biomineralization, easy tailoring of the microstructure, hydrophilicity, pore size, porosity, mechanical characteristics and degradability. They also have disadvantages, such as immunogenic and pathogenic impurities, poor replicability, need of crosslinking and an inferior loading-bearing capacity [35].

Hydrogels are based on hydrophilic polymers (natural, synthetic or hybrid) with a hydrophilic nature, with a high water content and permeability being substrates for supporting cell growth, which promotes osteogenesis, calcium biomineralization and angiogenesis [35].

*Four-dimensional biomaterials* are a new smart generation exhibiting a dynamic self-remodeling capability and a tunable stimuli responsiveness. They may contain hydrogels, bioceramics, piezoelectric materials, etc. that undergo self-transformation of shape and functionality after stimuli exposure [35].

The biomaterial/scaffold properties, such as pore size, stiffness, scaffold composition, surface topography, surface functional groups, surface wettability and degradation product, are crucial for their effects on cellular behavior being taken into consideration for the appropriate selection of a certain biomaterial-based scaffold [41]. The techniques applied for manufacturing the bone tissue regeneration scaffolds are freeze-drying, electrospinning, 3D printing (selective laser sintering, stereolithography, fused deposition modeling), solvent casting, sol-gel, gas foaming and particulate leaching [37]. Recently, a new 3D bioprinting technology that deposits living cells, extracellular matrices and biomaterials (inkjet-based bioprinting, extrusion-based bioprinting, laser-based bioprinting) has emerged [42].

Natural biomaterials have been widely applied in tissue regeneration due to their various advantages, such as biological and chemical similarity to natural tissues, biocompatibility, biodegradability, biological activity, low cytotoxicity, good cost-efficiency and availability. From this class, edible materials have attracted increasing interest due to some of their remarkable properties: enhancement of cell attachment, proliferation and migration and antibacterial, anti-inflammatory and antioxidant properties [43]. Edible materials, including natural polysaccharides (chitin, chitosan, hyaluronan, alginate, etc.), phenolic compounds (coumarin, phenolic acid, anthocyanin, lignin, tannic acid, etc.), and proteins (collagen, gelatin, silk fibroin, etc.) from plants, animals or other organisms, are a good source of nutrition for humans [43]. They can be employed as scaffolds or provide osteogenesis-stimulative substances for bone regeneration [43].

The inclusion of some plants, plant extracts or phytochemical compounds in the daily diet and in the biomaterials-based scaffold composition can improve the bone regeneration process. Phytochemical compounds from the classes of anthocyanins, phenols and flavonoids presented a beneficial and stimulating effect in the bone regeneration process.

Thus, the purpose of this review is to summarize the information regarding the implications of phytochemical extracts and their mechanism of action in the bone regeneration process.

## 2. Research Methodology

The studies considered were selected using the PRISMA 2020 flow diagram according to Page et al., 2021. The steps and selection criteria, followed by the number of the studies used for our review, are shown in Figure 1. Databases such as PubMed, Scopus, Science Direct, Elsevier, Google Scholar and Google Patents were accessed to search the literature. The Medical Subject Headings keywords included in the search were “bone regeneration”, “bone markers”, “bone signaling pathways”, “bone metabolism”, “bone pathology”, “plant extract”, “plant bone regeneration”, “natural compounds bone regeneration”, “elagic acid bone” and “bone regeneration materials”.

All the information systematized in the tables was obtained from research articles (in vivo/in vitro studies) and reviews between 2012 and 2022 (from the past 10 years). A Prisma flow-diagram was used to describe how to select the studies and articles included in the review, as shown in Figure 1 [44,45].

Studies published in languages other than English were excluded. A total of 143 studies were selected and included in this review.

## 3. Plant Extracts and Phytochemical Compounds with a Positive Effect on the Bone Regeneration Process

### 3.1. Classes of Phytochemical Compounds Involved in the Bone Regeneration Process

Natural products exhibit a wide range of modulatory effects on various pathways involved in bone regeneration (osteoclastogenesis inhibition, bone anabolism and bone resorption), such as NF-κB signaling pathways, MAPKs signaling pathways, Akt signaling pathways, calcium ion (Ca^2+^) signaling pathway, ROS-mediated effects and inflammatory mediator genes (Figure 2) [46,47]. Some phytochemical compounds have been effective in improving bone regeneration and preventing/treating osteoporosis and osteoarthritis by enhancing some mechanisms, such as mineral turnover, bone mineral density, inhibition of bone loss, increase in calcium and vitamin D3 and prevention of inflammation and oxidative stress [48].

The phytocompounds which have proven their potential in orthopedic applications belong to the following classes: flavonoids, tannins, polyphenols, anthocyanins, terpenoids, polysaccharides, alkaloids and others (Figure 2) [46,47,49]. Many researchers have pointed out that a certain activity is due to the complex composition of the plant product (including interactions such as the synergistic, additive, or antagonistic effect) rather than to a single compound, solvent extraction and doses [50].

### 3.2. Phytocompounds Used in the Bone Regeneration Process—State of the Art

Currently, many conventional treatments are known whose mechanism of action is to stop the reabsorption (anti-resorptive drugs) or anabolic drugs, but these are also known for their characteristic adverse effects following their long-term administration.

The treatment of osteoporosis can be carried out with the medicinal agents from the mentioned classes individually or as a combined treatment. A classic treatment against the loss of bone density is the combined therapy of anabolic and antiresorptive drugs, such as retiparatide, romosozumab and bisphosphonates, selective modulators of estrogen receptors, and the continuous administration of calcium combined with vitamin D. However, the demonstrated adverse effects suggest that these classical treatments should be administered with caution and not on a long-term basis. However, bone diseases such as osteoporosis and osteoarthritis are chronic diseases, so this premise is not valid.

Some of the adverse effects in the case of the long-term administration of the drug classes involved in conventional treatment are the occurrence of thrombotic events; the occurrence of breast cancer; the occurrence of kidney diseases (especially with the continuous administration of calcium together with vitamin D); gastrointestinal diseases; and cardiovascular events.

Considering these facts, the treatment plan for bone diseases—such as osteoporosis—brings to the fore natural molecules, specifically phytochemicals. These are of major interest for the treatment and stimulation of bone regeneration, and they do not cause any adverse effects.

According to the mechanism of action, natural compounds can be divided into three broad classes: compounds with pro-estrogenic activity, compounds with antioxidant and anti-inflammatory properties and modulatory compounds of bone regeneration pathways.

Estrogen has multiple effects on bone metabolism: Through its action on osteocytes, it inhibits bone remodeling; it also inhibits bone resorption through direct action on osteoclasts [51].

Oxidative stress disturbs the balance of bone metabolism, determining the apoptosis of osteocytes and osteoblasts and favoring the cell proliferation of osteoclasts. This causes bone destruction [52]. At the bone tissue level, oxidative stress favors postmenopausal, diabetic and glucocorticoid osteoporosis. Under the action of oxidative stress at the intracellular level, the mitochondria are deformed, which leads to the disruption of cellular metabolism and even to apoptosis [53].

Prostaglandin E2 (PGE2) is produced by both osteoblasts and osteoclasts in the initial phases of the bone healing process through a reaction catalyzed by the enzyme cyclooxygenase (COX). At the bone fracture, it promotes angiogenesis and increases the number of osteoclasts, which promotes osteolysis and increases the differentiation of osteoblasts, thus promoting osteogenesis. Both traditional non-steroidal anti-inflammatory drugs and selective COX2 inhibitors inhibit bone formation due to their effect on prostaglandins [54].

Proinflammatory cytokines regulate the inflammatory process of bones. They regulate both bone formation and bone resorption, thus altering bone homeostasis. Proinflammatory cytokines (TNFα, IL-1, and IL-17) cause osteoclast activation, which explains the increased bone loss during inflammation. Other cytokines, such as IL-12, IL-18, IL-33 and IFN, are suppressors of osteoclast differentiation and thus inhibit bone loss. The presence of certain cytokines in bone tissue can influence osteolysis [55].

Bone regeneration involves a large number of small molecules, transmission and signaling pathways, growth factors and physicochemical stimuli from the extracellular matrix, which are often interconnected and overlapping [56]. The main signaling pathways with a role in the bone regeneration process are Wnt, TGF-b, MAPK, JNK and the Notch pathway. Small molecules act as activators or inhibitors of transcription factors, and through this they can regulate the process of bone formation [57].

Several plant-derived components obtained from plant extracts proved their ability to affect the proliferation and differentiation potential of MSCs having as targets various signal transduction pathways demonstrating an osteopromotive role in bone regeneration [50,58,59,60]. The mechanism of action of some bioactive compounds are highlighted in Table 1.

## 4. The Innovative Administration and Application of Plant Extracts in the Process of Bone Regeneration

The conventional administration of treatments in bone diseases (oral, systemic administration) presents multiple disadvantages such as the low bioavailability of the administered medicinal substances or of the administered phytochemical compounds, the occurrence of side effects at the gastric and intestinal levels, low absorption and the need to increase the dose of administration [119,120,121].

Taking into account these inconveniences, traditional medicine is considered to be outdated with regards to bone diseases; therefore, the attention has been recently directed toward the targeted administration of natural compounds in nanometric form [122].

An advantage of using nanomedicine is the possibility of incorporating medicinal substances, phytochemicals or a mixture of two in different materials (biomaterials) that are compatible with bone tissue [121,122,123]. Moreover, because of their increased compatibility with the human body, the bioavailability of the substances administered through this route is increased, thus allowing a decrease in the doses administered [124,125,126].

The administration can be sustained at the level of the therapeutic plateau, and thus the number of administrations per day can be decreased. Increased compliance with the applied treatment can be thus gained [124]. The phytochemical compounds included in such systems are also protected from the factors that destroy their therapeutic activity, such as light, temperature and pH [123,124,125].

The nanometric systems also allow the controlled and targeted release of the contained compounds. The effectiveness of the treatment is much higher than that of conventional treatments [126,127,128]. Additionally, these systems allow local administration, at the bone level, which avoids the adverse gastric and intestinal effects that are encountered using conventional medication [129,130,131,132,133].

Some of these novel transport systems are presented in Figure 3.

New treatment opportunities and solutions are envisioned by combining herbal medicines having osteogenic, antitumor, antimicrobial, and anti-inflammatory properties with advanced materials for bone tissue engineering. Plant derived compounds have been included or added in the composition of orthopedic biomaterials (metallic, ceramic and polymeric matrix) in order to deliver the phytochemical substance at the bone site, leading to functionalized scaffolds with plant extracts applied in tissue engineering. A wide range of natural compounds have been proposed for incorporation within bone tissue engineering scaffolds in order to enhance bone growth, inhibit osteoclastic bone resorption, and prevent other bone-related complications [43,134,135,136]. Recently, an innovative concept has emerged in nanotechnology tissue engineering: combination and nanostructured scaffolds. These have attracted a huge interest due to their promising results in improving the bone healing process.

Essential oils (from different plant sources) and other compounds (metallic nanoparticles, zinc nitrate, copper sulphate, cobalt nitrate, etc.) have been used to treat polyurethane scaffolds, leading to the improved physical and biomedical properties of the designed scaffolds (stability, biocompatibility, bone mineralization, osteoblast cell adhesion, antimicrobial activity, etc.) [48]. Polyphenols, another class of natural compounds, have been incorporated in the composition of the bioactive scaffolds. They conferred unique structural and functional features: bio adhesion, antioxidation, anti-inflammatory and antibacterial properties, hydrophilicity, self-healing and biocompatibility (promoting bone regeneration). The approaches addressed for the fabrication of polyphenol-based scaffolds have included coating on the polymer scaffold and grafting or blending into biopolymers. They presented different morphologies, such as hydrogels (3D cross-linked networks), films (2D materials) and nanofibers (1D materials) [137].

Allium cepa extract, chitosan and poly (DL-lactic-co-glycolic) acid have been employed for the synthesis of a 3D matrix with a porous morphology (50–100 µm), allowing its surface mineralization to have a uniform hydroxyapatite layer [138]. Cucurbitacin B was incorporated into a biomaterial scaffold based on a poly (lactidecoglycolide) and β-tricalcium phosphate from where it was linearly released, showing enhanced neovascularization (via VEGFR-related signaling pathways) and bone regeneration (via higher bone mineral density, bone volume and number of trabeculae). The composite presented a bio-mimic structure with a pore between 16 and 466 nm and improved mechanical properties [114].

Aloe vera gel has been incubated on the surface of poly (3-hydroxybutyrate-co-3-hydroxyvalerate) nanofibers, generating a scaffold with promising osteoinductive potential (higher amounts of alkaline phosphatase activity, mineralization, and bone-related gene and protein expression were recorded) [139]. In another study, polycaprolactone/aloe vera/silk fibroin nanofibrous scaffolds were synthesized, followed by hydroxyapatite deposition. The result consisted of in biomimetic scaffolds with increased cell proliferation, osteogenic marker expression, osteogenic differentiation and mineralization [140].

A lipid-based self-nano emulsifying drug delivery system (100–180 nm) containing *Cassia occidentalis* L. butanolic extract prevented the downregulation of miR29a (Runx2), miR17 and miR20a (RANKL), induced by methyl prednisone and proving osteogenic and anti-resorptive mechanisms. The anti-inflammatory effect of the glucocorticoid was not affected, whereas the induced sarcopenia and muscle atrophy were counteracted [92]. Linum usitatissimum extract rich in phenolic compounds and flavonoids was incorporated in a composite hydrogel based on alginate and nano-hydroxyapatite which presented a porous structure (100–200 µm) and sustainedly released the natural compound. This composite demonstrated hemocompatibility, antioxidant activity and cell proliferation, thus promoting bone regeneration [141].

Fruits (grape seed, pomegranate peel, jabuticaba peel) extracts have been used as crosslinkers for anionic collagen, which together with nanohydroxyapatite formed a smart scaffold with a pore size (17–230 nm) appropriate for bone growth [142]. Genipin, a natural compound from gardenia fruits, demonstrated a promising cross-linking ability for natural biopolymers (collagen, gelatin, and chitosan) and a promising biosafety profile. The surface of several scaffolds was coated with genipin-crosslinked hydrogel releasing osteogenic factors, acting as an anti-infection agent or presenting a water-absorbing function [143].

Epigallocatechin gallate has been employed to coat poly (L-lactic acid) nanofibers to enhance hydrophilicity and stem cell adhesion, with the flavonoid serving as a protective agent of external oxidative stress for the stem cells. The plant flavonoid-based scaffold promoted the osteogenic differentiation of adipose-derived stem cells and reduced the osteoclastic maturation of the murine macrophages [101]. In another study, epigallo catechin gallate represented a promising tool to chemically modify gelatin sponges influencing surface properties (hydrophilicity and negative zeta potential). This functionalization enhanced cell adhesion and calcium phosphate precipitation, inducing superior bone formation in vivo [102]. In other study, gellan gelum hydrogels loaded with alkaline phosphatase were enriched with 5 types of gallotannins (three tannic acids with differing molecular weight, pentagalloyl glucose, and a gallotannin-rich extract from mango kernel (*Mangifera indica* L.). These preparations promoted the mineral formation (the dry mass percentage values were increased) due to the interactions between the ALP and gallotannins dependent on the medium [104].

Forskolin, a triterpenoid from *Coleus forskohlii,* was loaded into halloysite nanotubes, which were then used as a dopant for the modification biopolymer scaffold (based on gelatin, chitosan, agarose). This resulted in a new osteoconductive smart polymeric scaffold. The forskolin-loaded halloysite nanotubes acted as a filler that increased the composite volume and filled the voids, therefore facilitating the cell attachment. The mechanical properties of the scaffold together with the chemical signal of forskolin (cyclic adenosine monophosphate signaling activation in stem cells) had a synergic effect in promoting the osteodifferentiation of mesenchymal stem cells [103].

Plant-derived nanoparticles from potato containing rhamnogalacturonan-I (with relatively higher amount of gallactose) have been used for poly (L-lactide-co-ɛ-caprolactone) scaffolds functionalization revealing a downregulation of pro-inflammatory gene markers and promotion of osteogenic markers [79]. The same plant-derived pectin, rhamnogalacturonan-I, has been employed for the nanocoating of im-plants influencing osteoblast proliferation, mineralization and gene expression (Runt-related transcription factor 2 (Runx2), alkaline phosphate (ALP), osteocalcin (Bglap), α-1 type I collagen (Col1a1), receptor activator of NF-κB ligand (RANKL)) [80]. A plant-derived phenolic compound, sinapic acid, was used for the synthesis of sinapic-acid-loaded chitosan nanoparticles (100-115 nm), which were incorporated into polycaprolactone fibers (around 350 nm), resulting in a scaffold that promoted osteoblast differentiation in vitro and bone formation in vivo (via TGF-β1/BMP/Smads/Runx2 signaling pathways) due to the sustained release of sinapic acid [112].

## 5. Conclusions

Understanding the structure of bone tissue and the mechanisms of bone formation is crucial in the development of new methods of treatment for bone defects. The purpose of this work was to summarize the main cells, biomarkers and signaling pathways involved in bone metabolism. The phytochemical compounds with huge potential in bone regeneration were highlighted according to their activity and mechanisms of action included in three broad classes: natural compounds with pro-estrogenic activity, natural compounds with antioxidant and anti-inflammatory properties and modulatory compounds of bone signaling pathways. Recently, the targeted administration of natural compounds in nanometric forms for bone regeneration has been in high demand due to their considerable advantages. Another trend of nanomedicine applied in orthopedic field consists in the combination of plant-derived compounds with nanostructured biomaterials, resulting in functionalized scaffolds that are crucial for bone tissue engineering and that have shown promising results for the improvement of the bone healing process.

## Figures and Tables

**Figure 1 plants-12-02055-f001:**
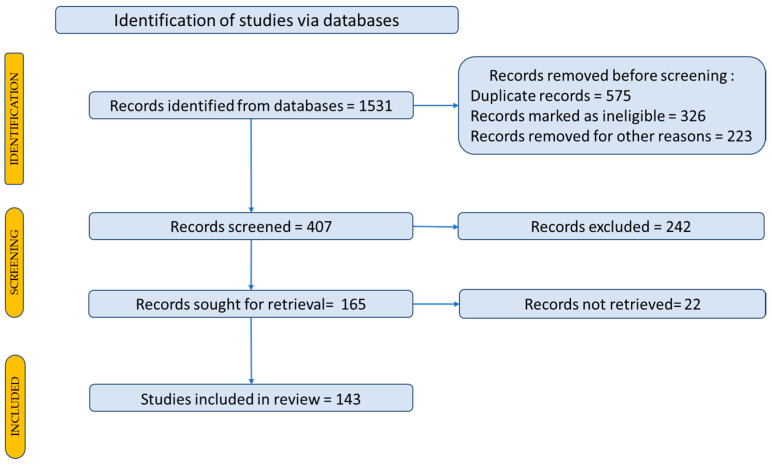
Prisma flow-diagram for description of the selection process of the bibliographic sources.

**Figure 2 plants-12-02055-f002:**
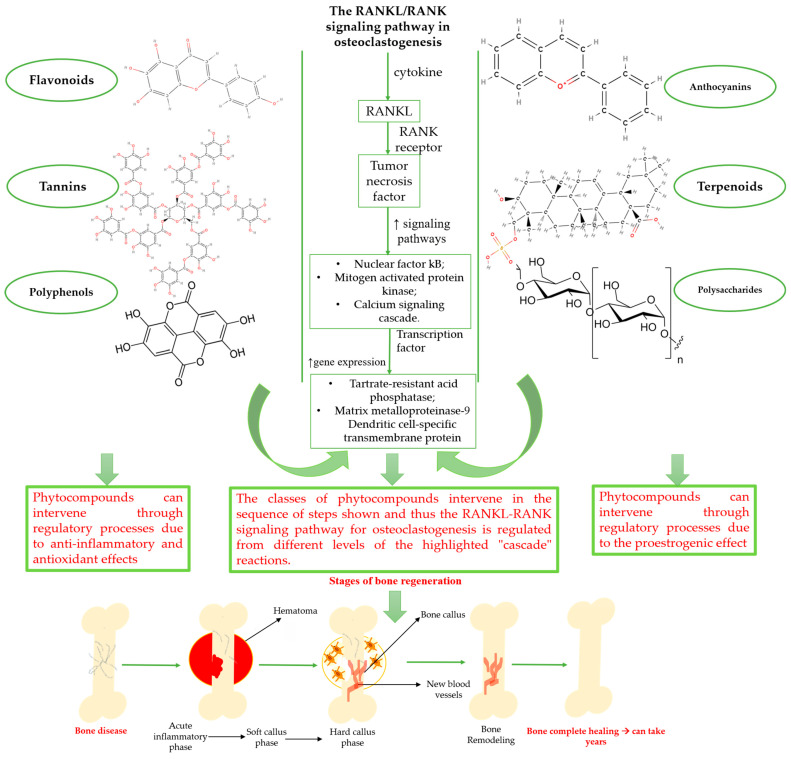
The main classes of phytocompounds and the main regulatory mechanisms involved in the bone regeneration process, the stages of bone density restoration in different pathologies.

**Figure 3 plants-12-02055-f003:**
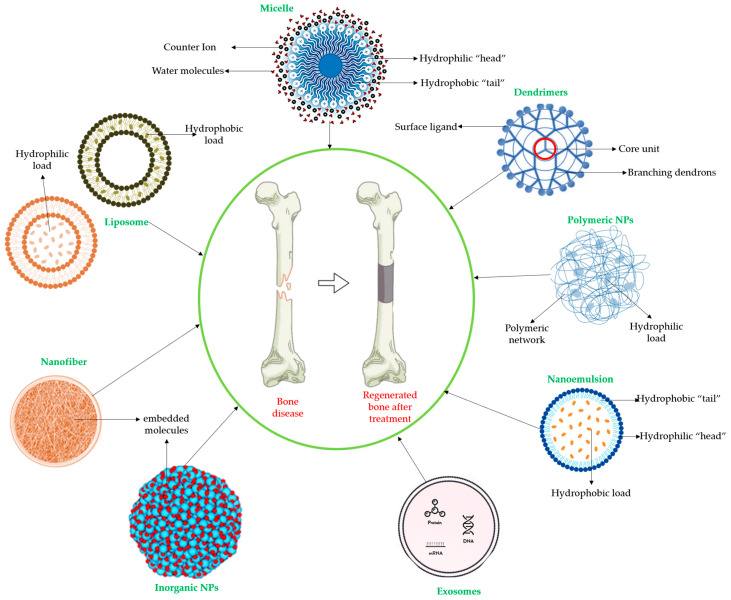
The diversity of nanometric systems used for the innovative application of phytocompounds for the purpose of bone regeneration.

**Table 1 plants-12-02055-t001:** Phytochemical compounds involved in bone regeneration process including their mechanism of action and natural sources.

Compounds	Type of Activity	Mechanism of Action	Extract Source	In vivo/in vitro Studies	Ref.
**Genistein** 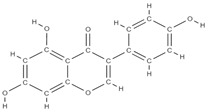	Proestrogenic activity	↑ alkaline phosphatase level ↓ urinary excretion of calcium and phosphate, → serum concentration at the appropriate normal level	*Erythrina variegate*	In vitro (mesenchymal stem cells)	[61]
**Daidzein** 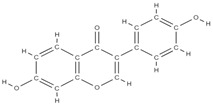		↑ osteoclast apoptosis through the mediation of estrogen receptors ↓ the loss of bone density activates tyrosine phosphatase → ↓ membrane depolarization producing changes in intracellular Ca^2+^		In vivo (rats)	[62]
**Icariin** 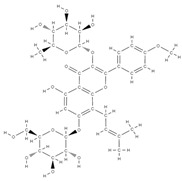		↓ bone loss in the median bone area by regulating the ratio between osteoprotegerin and RANKL, which are key mediators of osteoclast genesis. ↑ proliferation, differentiation of osteoblasts, bone mineralization ↓ cell apoptosis direct osteoblast stimulation: activation of the bone morphogenetic protein (BMP) cascade through (promoting Runx2/Cbfa1 expression and the production of BMP-4, BMP-2, and SMAD4 and nitrous oxide release; high levels of ALP suppression of p38 and JNK pathways in the osteoclasts, ↓ release of prostaglandin E2 by osteoblasts => inhibition of osteoclast differentiation	*Epimedium*	In vivo (rats)	[63] [64]
**Dioscin** **e** 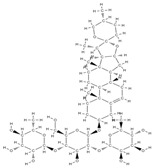		↑the proliferation of bone tissue ↓ cell apoptosis by mediating signaling pathways ↓RANKL expression ↓osteoprotegerin/RANKL → inhibits bone reabsorption	*Dioscoreaceae family*	In vivo (mice)	[65]
**Kaempferol** 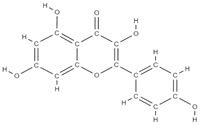		↑ osteoprotegerin and ↓ RANKL expression → osteoclastogenesis decreases ↑ antiapoptotic expression maintaining bone mass, microarchitecture, and bone strength of the trabecular bones	*Ginkgo biloba* *Camellia sinensis*	In vivo (rats) In vivo (rats) In vivo (rats)	[66,67,68] [69]
**Quercetin** 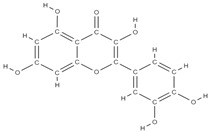		↑ the proliferation of bone tissue ↓osteoprotegerin/RANKL → inhibits bone reabsorption		In vitro (mesenchymal stem cells) In vitro (periodontal ligament cells)	[61,69]
**Ginkgolic acid** 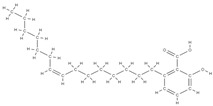		↑ proliferation, differentiation of osteoblasts, bone mineralization	*Ginkgo biloba*	In vitro (mesenchymal stem cells)	[61]
**Caviunin** 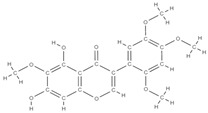		stimulates BMP-2/Wnt-βcatenin pathway	*Dalbergia sissoo*	In vivo (rats)	[70]
**Acteoside** 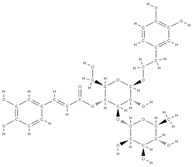	Antioxidant and anti-inflammatory effect	↓ the level of pro-inflammatory cytokines such as TNF-α and IL-6, ↓ the differentiation of osteoclasts by reducing free radicals and fighting oxidative stress ↑ cell proliferation ↓ bone demineralization	*Verbascum sp.* *Cistanche sp.*	In vivo (rats)	[71]
**Curcumin** 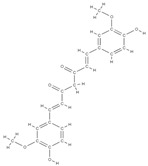		↓the level of inflammation by decreasing the inflammatory cytokines TNF-a and IL-6 ↓bone loss and demineralization, inhibiting osteoclastogenesis ↑the level of alkaline phosphatase, which leads to an increase in the mineralization process interaction with transcription and growth factors, protein kinases, cytokines and enzymes => apoptosis of cancer cell	*Curcuma longa*	In vitro (osteosarcoma cells) In vitro (human osteosarcoma cells)	[72] [73]
**Resveratrol** 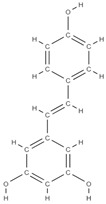		↓the level of free radicals from the bone level, neutralizing them ↓bone loss inhibits osteoclastogenesis and the RANKL marker influences the response of estrogen receptors to oxidative stress factors ↑bone differentiation → ↑ bone density ↑the level of morphogenetic protein at the bone level ↓decreases the level of alkaline phosphatase ↓the level of osteocalcin. allows mass production of MSCs; mRNA levels of RUNX2, Collagen Type I Alpha 1 (COL1A1), PPARγ, Adiponectin (APN) were highly expressed, ↑ SIRT1 and SOX2 levels	-	In vivo (rats) In vivo (rats) In vitro (mesenchymal stem cells)	[74,75,76,77] [77]
**Gomisin** 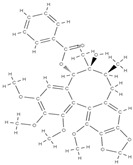 **Schisandrin C** 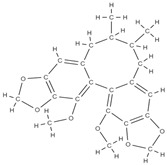		down-regulation of inflammatory molecules, ROS, and up-regulation of antioxidant molecules	*Schisandra chinensis*	In vitro (murine macrophage, myoblasts, human diploid fibroblasts, bone marrow macrophages, osteoblasts) In vivo (rats)	[78]
**Rhamnogalacturonan-I** 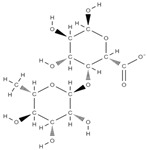		↓ intracellular accumulation of galectin-3 down-regulation of RANKL, TNFα, IL-6, and IL-1β	*Solanum tuberosum*	In vitro (neutrophils and macrophages; osteoblasts) In vivo (rats)	[79,80]
**Acemannan** 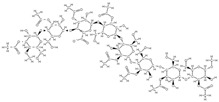		tissue regeneration, cell proliferation, extracellular matrix synthesis, mineralization. ↑ expression of growth factors; stimulation of bone cementum and periodontal ligament regeneration; induction of bone formation, osteoblast proliferation and differentiation	*Aloe vera*	In vitro (mesenchymal stem cells) In vivo (rats)	[81]
**Ellagic acid** 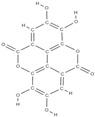 **Caffeic acid** 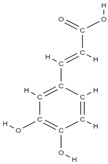		- inhibition of iNOS, COX-2, NO, TNF-α, PGE2 and IL-6 - down-regulation of IL-1β-stimulated matrix metalloproteinase-13 and thrombospondin motifs 5 - up-regulation of collagen of type II and aggrecan - suppression of NF-κB signaling - ↓ chitinase-3-like protein-1, IL-1β, NF-κB, caspase-3; lipid peroxides, NO - ↑ reduced glutathione		In vivo (mice) In vitro (human chondrocytes) In vivo (rats)	[82] [83]
**Ginsenoside** 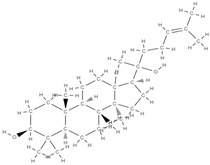	Modulatory compounds of bone regeneration pathways	↑ calcium absorption at the intestinal level → thus prevents bone loss ↑ the level of trabecular calcium ↓ C-terminal telopeptide of type I collagen → ↓ resistance to tartrate acid phosphatase at the femoral level	*Orchidaceae family*	In vitro (mesenchymal stem cells)	[61,84]
**Berberine** 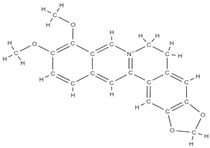		↓ bone loss by preventing decalcification and demineralization inhibits osteoclastogenesis suppresses the activity of the markers involved in the differentiation of acid phosphatase-resistant tartrate bone cells and cathepsin K ↓ the differentiation rate of osteoclasts restore downregulation of osteogenesis-related genes expression; ↑ expression of osteogenesis-related genes such as OSX, COLⅠ, ALP, OCN and OPN ↑ total β-catenin and nuclear β-catenin; activation of the Wnt/β-catenin signaling pathway	*Coptis species.* *Berberis species.* *Coptidis Rhizoma, Coptis chinensis, Coptis teeta.*	In vitro (mesenchymal ctem cells) In vitro (osteoblast and osteoclast) In vitro (mesenchymal stem cells)	[85,86,87,88] [87]
**Apigenin** 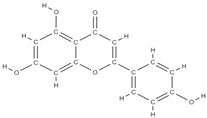		↑ the proliferation capacity of osteoblasts inhibits decalcification and osteoclastogenesis modulates intracellular signals → ↓bone loss induced by estrogen hormones ↓ the level of bone inflammation. ↑ mRNA levels of osteogenic genes BMP-2, Runx2 and COL1 downregulation of miR29a, miR17 and miR20a	*Olea europaea.* *Cassia occidentalis*	In vivo (rats) In vitro (osteoblasts) In vitro (osteoblasts) In vivo (rats)	[89,90,91,92]
**Chlorogenic acid** 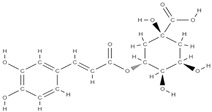		↑ the level of favorable markers for bone formation ↑ the level of bone morphogenetic protein →↑ the activity of osteoblasts ↓ the level of pro-inflammatory factors ↑ the level of glutathione peroxidase →strong antioxidant effect ↑ the serum activity of alkaline phosphatase, osteoprotegerin ↓ the production of RANKL decreases	*Prunus domestica L.*	In vivo (rats) In vivo (rats)	[93,94]
**Aesculetin** 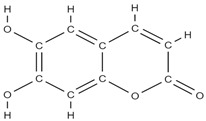		↑ expression of bone morphogenetic protein-2, collagen type 1, osteoprotegerin; ALP activation; transcription of Runt-related transcription factor 2; induction of: non-collagenous proteins of bone sialoprotein II, osteopontin, osteocalcin, and osteonectin, of annexin V and PHOSPHO 1. ↑ the production of thrombospondin-1 and tenascin C	-	In vitro (osteoblasts)	[95]
**Acemannan** 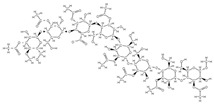		↑ mRNA expression of bone morphogenetic protein 2 ↑ mineral deposition	*Aloe vera*	In vivo (volunteers)	[96]
**Antihemorrhagic plant extract**		↑ osteoblastic activity and new bone formation; ↑ osteonectin and osteopontin expression ↓ inflammatory cell infiltration, vascular dilatation and hemorrhage	*Glycyrrhiza glabra, Vitis vinifera, Alpinia officinarum Urtica dioica, Thymus vulgaris*	In vivo (rats)	[97]
**Withaferin A** 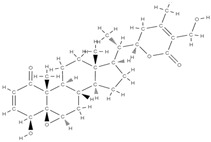		↑ expression of osteoblast-specific transcription factor and mineralizing genes, osteoblast survival, ↓ inflammatory cytokines.	*Withania somnifera*	In vitro (osteoblasts) In vivo (mice, rats)	[98]
**Ecdysterone** 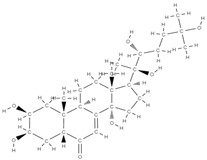		↑ gene expression of the BMP-2/Smad/Runx2/Osterix signaling pathway, stimulates MC3T3-E1 cell proliferation		In vitro (osteoblasts) In vivo (rats)	[99]
**Echinacoside** 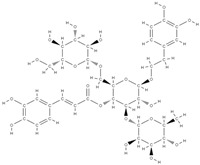		↑ the uterine weight and serum E2 levels, ↓ body weight and hydroxyproline serum levels	*Cistanche tubulosa*	In vivo (rats)	[100]
**Epigallocatechin gallate** 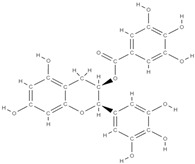		activation of β-catenin of the Wnt signaling pathway ↑ expression of osteogenic genes, ALP activity, and mineralization in bone marrow-derived mesenchymal stem cells	*Grean tea*	In vitro (adipose-derived stem cells, dedifferentiated fat cells) In vivo (mice, rats)	[101,102]
**Essential oils**		blocking nuclear factor kappa B, p38, and c-Jun N-terminal kinase signaling ↓ production of nitric oxide in RAW264.7 cells, inhibited EAhy926 cell proliferation ↑ serum C-telopeptide collagen type I and osteocalcin ↑ plasma calcium and vitamin D3, bone mineral-density Prevention of inflammation and oxidative stress	*Hypericum perforatum;* *Cinnamomum burmanini;* *Thymus vulgari;* *Rosmarinus officinalis.* *Populus alba;*	In vitro (macrophages, fibroblasts, osteoblasts) In vivo (rats, mice)	[48]
**Forskolin** 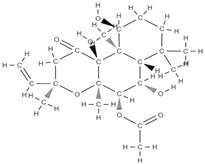		activation of cyclic adenosine monophosphate (c-AMP) signalling in stem cells	*Coleus forskohlii*	In vitro (mesenchymal stem cells)	[103]
**Gallotannin** 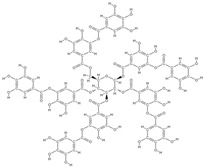		interaction with ALP growth of Saos-2 cells	*Mangifera indica L.*	In vitro (osteoblasts)	[104]
**Ursolic acid** 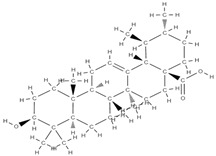		↑ trabecular parameters (BV/TV, Tb.Th and conn.D) ↓ SMI ↑ALP activity, osteogenic genes (Runx2, BMP-2, type 1 Col1 and Wnt3a) stimulates Wnt/β-catenin signalling osteoblast differentiation (activation of mitochondrial respiration)	*Psidium guajava*	In vitro (osteoblasts) In vivo (rats)	[105]
**Malvidin** 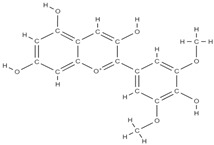 **Cyanidin** 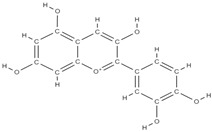 **Delphinidin** 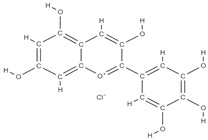		inhibition of MSC adipogenesis and downregulation of FABP4 and adiponectin genes. ↑ accumulation of calcium deposits upregulation of osteocyte-specific gene BMP-2 and Runx-2 expression	*Berries*	In vitro (mesenchymal stem cells)	[106]
**Rutin** 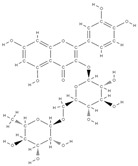		activation of Wnt/b-Catenin Signaling ↑ activity of ALP, Runx2, osterix, osteocalcin, bone morphogenetic protein 2, Wnt3a, and b-catenin	*Morinda citrifolia (Noni)*	In vitro (murine myoblast cell line, human periodontal ligament cells) In vivo (rats)	[107,108,109]
**Rhamnogalacturonan-I** 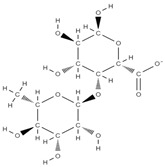		↓ intracellular accumulation of galectin-3 up-regulation of collagen type I alpha 1 (COL-Iα1), osteocalcin, sialoprotein. down-regulation of RANKL, TNFα, IL-6, and IL-1β	*Solanum tuberosum*	In vitro (neutrophils and macrophages; osteoblasts) In vivo (rats)	[79,80]
**Crocin** 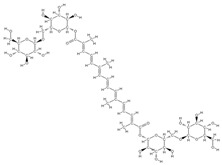 **Crocetin** 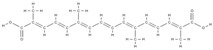		↑ ALP activity and ALP mRNA expression in MSCs	*Crocus sativus L.*	In vitro (mesenchymal stem cells)	[110]
**Sinapic acid** 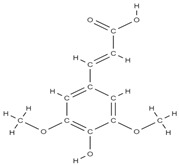		activation of TGF-β1/BMP/Smads/Runx2 signaling pathways => osteoblast differentiation	*Cynanchi atrati*	In vitro (macrophags) In vitro (mesenchymal stem cells) In vivo (rats)	[111,112]
**Beta** **Ecdysone** 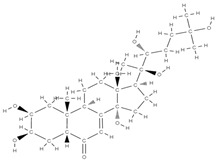		↑ collagen deposition, ↑ levels of osteocalcin, ↑ expression of osteogenic genes	*Tinospora cordifolia*	In vitro (osteoblasts, macrophages) In vivo (rats)	[113]
**Cucurbitacin B** 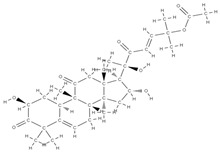		↑ expression of ALP and OPN genes, mineralization up-regulation of VEGFR2 and VEGFR-related signaling pathways (induction of angiogenesis)	*Cucurbitaceae family plants*	In vitro (mesenchymal stem cells) In vivo (rats)	[114]
**Polysaccharides**		hematopoiesis protection: ↓ myeloid cells within tumor-infiltrating immune cells Inhibition of hematopoietic cell expansion in the spleen ↑ HSPCs (hematopoietic stem and progenitor cells) and common lymphoid progenitors in the bone marrow	*Polygonatum sibiricum*	In vivo (mice)	[115]
**Ellagic acid** 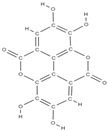 **Ellagic acid and** **Sennoside B** 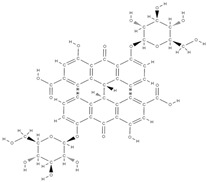		↑ number of osteoblasts and expression of OCN and OPG ↓ number of osteoclasts and the expression of RANKL - repression of c-Jun expression at the mRNA level		In vivo (rats) In vitro (human osteosarcoma cells)	[116] [117]
**Ellagic acid and hydroxyapatite**		↑ in the expression of FGF-2, VEGF and ALP ↑ IL-10, BMP-4 and OPN ↓ TNF-α and increasing the expression of		In vivo (rats)	[118,119]
**Melibiose** 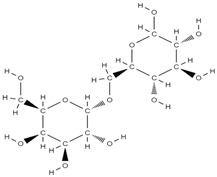 **Methylophiopogonanone A** 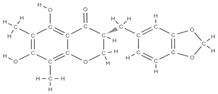 **Tubuloside A** 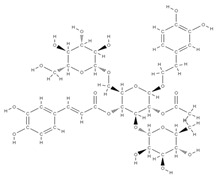		↑ expression of ALP, osteocalcin, osterin, osteoprotegerin, and autophagy marker proteins activation of BMP2/Smad/Runx2 and Wnt/β-catenin signaling	*Juglans regia*	In vitro (mesenchymal stem cells)	[120]

RANK: receptor activator of nuclear factor kappa B; RANKL: receptor activator of nuclear factor kappa B ligand; BMP: bone morphogenetic protein; ALP: alkaline phosphatase; MSCs: Mesenchymal stem cells; APN: Adiponectin; COL-Iα1: collagen type I alpha 1; FABP4: fatty acid binding protein 4; RUNX2: runx-related transcription factor 2; ROS: reactive oxygen species; IL: interleukin; TGF-β: Transforming growth factor-beta; VEGF: vascular endothelial growth factor; JNK: *c*-jun-*N*-terminal kinase; E2: endogenous 17-β-estradiol; Wnt: the Wingless-type MMTV integration site family; PPARγ2: peroxisome proliferator-activated receptor γ2; SIRT 1: sirtuin 1; OSX: Osterix; OPN: osteopontin; SMAD: Small mother against decapentaplegic; iNOS: inducible nitric oxide synthase, COX-2: cyclooxygenase-2, NO: Nitric oxide, PGE2: prostaglandin E2, NF-κB: nuclear factor kappa B, OCN: osteocalcin protein, OPG: osteoprotegerin, OPN: osteopontin. Chemical structures were obtained from the site https://molview.org/ accessed on 17 May 2023.

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
