# Peer review of "Phytochemical Compounds Involved in the Bone Regeneration Process and Their Innovative Administration: A Systematic Review"

_plants, 2023, doi:10.3390/plants12102055_

Round 1

Reviewer 1 Report

The current study titled “Phytochemical compounds involved in the bone regeneration process and their innovative administration: a systematic review” Ref: 2376389, constitute a short review with brief knowledge for the titled subject.

Chemical structure of the compounds mentioned in Table 1 should be inserted beside the mentioned name.

Author Response

We want to thank to the reviewer for your kind suggestions contributing to a considerable improvement of the manuscript. We have responded to the reviewers' comments, meanwhile changes have been made to the manuscript, written in red.

The current study titled “Phytochemical compounds involved in the bone regeneration process and their innovative administration: a systematic review” Ref: 2376389, constitute a short review with brief knowledge for the titled subject.

Chemical structure of the compounds mentioned in Table 1 should be inserted beside the mentioned name.

Correction was done according to your suggestion: the chemical structure of each compound was inserted below the mentioned name.

Reviewer 2 Report

This is an interesting review identifying the plethora of phytochemicals reported to influence skeletal cells and bone metabolism. 

General – This manuscript will need thorough proofreading and some reorganisation prior to publication. 

The comments below are not an exhaustive list but some specific examples that need attention:

Lines 38-39 – The opening statement needs revising; what do the authors mean by “great resistance” and “intense activity”?

Line 44 – It’s BMU NOT “MBU”, likewise line 45.

Line 54 – Revise the opening statement to: “….(PTH) is one of the most….regulators.”

Line 61 – Lower case “v” for vitamin D, likewise for the rest of that section.

Line 64 -  Lower case “c” for calcium.

Line 70 – Revise to “will be described.” – you need the correct tense. This applies to other areas of the manuscript.

Line 76 – Whilst osteocytes are indeed terminally differentiated osteoblasts they are not directly involved in bone mineralisation. Please revise, likewise line 78.

Lines 97-100 – The authors need to include active vitamin D3 as a key player in the regulation of osteoblast differentiation.

Lines 127-131 – The authors need to stress the importance of alkaline phosphatase through the clinical correlate of hypophosphatasia.

Lines 142 – 145 – RANKL promotes osteoclast differentiation and resorptive activity, please revise. Also, content duplication for RANKL/RANK/OPG – lines 160-164 – recommend this appearing once, decide where it should sit and revise accordingly.

Line 155 – Delete “cell lines” and just have osteoblasts as the last word.

Line 188 – Revise to: “have all three”

Lines 201-202 – in vitro and in vivo need to be italicised – check the remainder of the document.

Lines 228 – 233 – Would recommend restructuring, it’s too long. Start with the “pros” and have a second part dealing with the “cons”.

Line 231 – Revise to “immunogenic”.

Lines 257-259 – This is a bit nebulous and needs rewording/restructuring with appropriate citations to support the statements made.

Line 269 – “according to Page et al. 2021” will suffice here.

Line 270 – “Figure 1” in bold font, yet for line 278  “Figure 1” in regular font.

Line 302 – You already have a “Figure 1” – the Prisma flow-diagram.

Line 308 – Should be Figure 2 – the molecules depicted could be of better quality. It’s “Matrix metalloproteinase-9” NOT “metallopeptidase”. Overall this is not a very accessible figure.

Line 370 – Table 1 – for the sake of completeness, the identities of the cells/whole animal studies responding to the different phytochemicals should be identified for all.

Line 403 – Revise to “Figure 3”.

Line 480 – Revise to “composite”.

Line 489 – Revise to “implants”.

Line 491 – Revise to “alkaline phosphatase”.

Line 493 – Revise to “synthesis”.

Line 510 – Revise to “have shown”.

Line 511 – Revise to “the bone”.

Author Response

We want to thank to the reviewer for your kind suggestions contributing to a considerable improvement of the manuscript. We have responded to the reviewers' comments, meanwhile changes have been made to the manuscript, written in red.
